# Sequential *i*-GONAD: An Improved In Vivo Technique for CRISPR/Cas9-Based Genetic Manipulations in Mice

**DOI:** 10.3390/cells9030546

**Published:** 2020-02-26

**Authors:** Masahiro Sato, Rico Miyagasako, Shuji Takabayashi, Masato Ohtsuka, Izuho Hatada, Takuro Horii

**Affiliations:** 1Section of Gene Expression Regulation, Frontier Science Research Center, Kagoshima University, Kagoshima 890-8544, Japan; k4773775@kadai.jp; 2Laboratory Animal Facilities & Services, Preeminent Medical Photonics Education & Research Center, Hamamatsu University School of Medicine, 1-20-1 Handayama, Higashi-ku, Hamamatsu, Shizuoka 431-3192, Japan; shuji@hama-med.ac.jp; 3Department of Molecular Life Science, Division of Basic Medical Science and Molecular Medicine, Tokai University School of Medicine, Isehara, Kanagawa 259-1193, Japan; masato@is.icc.u-tokai.ac.jp; 4Center for Matrix Biology and Medicine, Graduate School of Medicine, Tokai University, Isehara, Kanagawa 259-1193, Japan; 5The Institute of Medical Sciences, Tokai University, Isehara, Kanagawa 259-1193, Japan; 6Laboratory of Genome Science, Biosignal Genome Resource Center, Institute for Molecular and Cellular Regulation, Gunma University, 3-39-15 Showa-machi, Maebashi 371-8512, Japan; hatada@gunma-u.ac.jp (I.H.); horii@gunma-u.ac.jp (T.H.)

**Keywords:** *i*-GONAD, knock-in, indels, genome-editing, oviducts, electroporation, Cre/*lox*P, in vivo gene delivery

## Abstract

Improved genome-editing via oviductal nucleic acid delivery (*i*-GONAD) is a technique capable of inducing genomic changes in preimplantation embryos (zygotes) present within the oviduct of a pregnant female. *i*-GONAD involves intraoviductal injection of a solution containing genome-editing components via a glass micropipette under a dissecting microscope, followed by in vivo electroporation using tweezer-type electrodes. *i*-GONAD does not involve ex vivo handling of embryos (isolation of zygotes, microinjection or electroporation of zygotes, and egg transfer of the treated embryos to the oviducts of a recipient female), which is required for in vitro genome-editing of zygotes. *i*-GONAD enables the generation of indels, knock-in (KI) of ~ 1 kb sequence of interest, and large deletion at a target locus. *i*-GONAD is usually performed on Day 0.7 of pregnancy, which corresponds to the late zygote stage. During the initial development of this technique, we performed *i*-GONAD on Days 1.4–1.5 (corresponding to the 2-cell stage). Theoretically, this means that at least two GONAD steps (on Day 0.7 and Day 1.4–1.5) must be performed. If this is practically demonstrated, it provides additional options for various clustered regularly interspaced palindrome repeats (CRISPR)/Caspase 9 (Cas9)-based genetic manipulations. For example, it is usually difficult to induce two independent indels at the target sites, which are located very close to each other, by simultaneous transfection of two guide RNAs and Cas9 protein. However, the sequential induction of indels at a target site may be possible when repeated *i*-GONAD is performed on different days. Furthermore, simultaneous introduction of two mutated *lox* sites (to which Cre recombinase bind) for making a floxed allele is reported to be difficult, as it often causes deletion of a sequence between the two gRNA target sites. However, differential KI of *lox* sites may be possible when repeated *i*-GONAD is performed on different days. In this study, we performed proof-of-principle experiments to demonstrate the feasibility of the proposed approach called “sequential *i*-GONAD (*si*-GONAD).”

## 1. Introduction

Gene editing technologies, such as the clustered regularly interspaced short palindrome repeats (CRISPR)/Caspase 9 (Cas9) system, have been frequently employed as convenient and efficient tools for performing gene modification in various biological systems [1,2]. Particularly, the generation of genome-edited mice by CRISPR/Cas9 has been considered important for analyzing the biological function of a gene of interest (GOI) and for the development of a disease mouse model. Genome-edited mice can be generated via microinjection of genome-editing components into the zygote or by subjecting the zygotes to electroporation (EP) in vitro in the presence of genome-editing components [3,4,5,6,7,8,9,10,11,12,13,14,15]. However, the microinjection technique requires expensive micromanipulator and skilled handling, while EP requires an expensive electroporator. Yoon et al. [16] and Mizuno et al. [17] demonstrated successful genome-editing of the target gene of mouse zygotes, by incubating the zona pellucida (ZP)-intact zygotes in the medium containing recombinant adeno-associated virus serotype 6 (rAAV-6) carrying genome-editing components or by subjecting the zygotes to EP in the presence of rAAV-6 carrying genome-editing components. All these procedures require ex vivo handling of embryos, which involves subjecting the zygotes derived from in vitro fertilization (IVF), or those freshly isolated from the pregnant females to genome-editing prior to egg transfer into the oviducts of recipient females. Furthermore, these techniques involve labor-intensive and time-consuming ex vivo handling of embryos.

In 2015, we first demonstrated the possibility of the production of a genome-edited mouse in situ [18]. They exposed the ovary/oviducts by making an incision in the skin of a pregnant female mouse (at Day 1.4 of pregnancy; corresponding to the 2-cell stage) under anesthesia, followed by inserting a glass micropipette through the oviductal wall. Next, a small amount of a solution (1–1.5 μL) containing Cas9 and guide RNA (gRNA) was injected into the oviducts with the aid of a mouthpiece-controlled micropipette under a dissecting microscope. Subsequently, in vivo EP was performed using tweezer-type electrodes. It is conceivable that the exogenous genome-editing components instilled within the oviductal lumen is incorporated into the 2-cell embryos and the genome is edited at the target locus. The molecular biology analysis of mid-gestational fetuses dissected from females subjected to surgery revealed that this technique successfully edited the genome with relatively high efficiency (approximately 29%). They named this technology “genome-editing via oviductal nucleic acids delivery (GONAD)”. We further improved this technology by injecting a solution containing Cas9/gRNA complex (called ribonucleoprotein, RNP) or single-stranded (ss) DNA donor (for knock-in (KI) experiment) into the oviducts of a pregnant female (at Day 0.7 of pregnancy: corresponding to the late 1-cell stage) [19]. This modification resulted in 97% of the embryos exhibiting insertion and deletion mutations (indels) in the target locus and approximately 50% of embryos containing KI alleles. We re-named this improved technology as improved genome-editing via oviductal nucleic acid delivery (“improved GONAD (*i*-GONAD)”). *i*-GONAD is also demonstrated to be an effective tool for in vivo genome-editing in rats [20,21].

In this study, we aimed to extend the applicability of *i*-GONAD technology. *i*-GONAD was performed twice with a 1-day interval between the two procedures. The first *i*-GONAD was performed on a pregnant female at Day 0.7 of pregnancy. One day later, the second *i*-GONAD was performed on the same female (at Day 1.7 of pregnancy; corresponding to the 2-cell stage) (Figure 1). This technology was called “sequential *i*-GONAD (*si*-GONAD)”. The detailed procedure for *si*-GONAD is schematically shown in the right panel of Figure 1. This technique enables the induction of two types of indels at two target sites that are located very close to each other. Furthermore, the sequential introduction of two mutated *lox* sites (to which Cre recombinase bind) at a target locus may be possible, which has been difficult as it often causes large deletion (LD) of a sequence, flanked by the sites to which the two *lox* sites are knocked-in [22].

In this study, we performed proof-of-principle experiments to demonstrate the feasibility of our proposed approach using two fluorescent dextrans (for checking whether an embryo can incorporate two types of substances through *si*-GONAD), two gRNAs targeted to mouse *GGTA1*, a gene coding for α-1,3-galactosyltransferase (α-GalT) capable of synthesizing α-Gal epitope, a cell-surface carbohydrate (for checking whether genome-editing is induced at the two sites, which are located close to each other, through *si*-GONAD), two gRNAs (targeted to intronic portions interposing exon 3 of *Mecp2* gene), and two single-stranded oligodeoxynucleotides (ssODNs) (containing mutated *lox* sites as donor DNA) (for checking whether KI of two *lox* sites occurs through *si*-GONAD).

## 2. Materials and Methods

### 2.1. Mouse and Superovulation Induction

According to our previous paper [19], *i*-GONAD is usually performed at approximately ~16:00 to 18:00 h from the day of the vaginal plug detection, which corresponds to the late 1-cell stage. However, this timing often causes technical inconvenience for researchers and technicians. We previously demonstrated that the timing of ovulation and subsequent fertilization by sperm can be shifted by changing the time of gonadotrophin administration to an earlier time point [23]. Thus, in this study, we used a modified regimen of superovulation suitable for *i*-GONAD experiments. Briefly, the adult B6C3F1 females (hybrids between C57BL/6 and C3H/He; 10–16 weeks of age; purchased from Kyudo Co. Ltd., Tosu, Saga, Japan) were intraperitoneally (IP) injected with 5 IU of pregnant mare’s serum gonadotropin (PMSG; eCG) at approximately 11:00 h, followed by administration of the same dose of human chorionic gonadotrophin (hCG) 48 h later. Immediately after the injection of hCG, the females were allowed to mate with the fertile males. The next morning (approximately 11:00 h; corresponding to Day 0.7 of pregnancy), the pregnant females, which have the vaginal plug, were subjected to *i*-GONAD. The day when the vaginal plug is recognized is defined as Day 0 of pregnancy.

The mice were maintained on a 12 h light/dark cycle (lights on from 07:00 h to 19:00 h) and were provided with food and water ad libitum. The experiments described were performed according to the guidelines of the Kagoshima University Committee on Recombinant DNA Security (No. 25035; dated 30 May 2018) and were based on the ‘Guide for the Care and Use of Laboratory Animals’ of the National Academy of Sciences, USA. Additionally, the experiments were approved by the Animal Care and Experimentation Committee of Kagoshima University (Sakuragaoka Campus) (No. MD18035; dated 25 September 2018). The experiments involving the in vivo transfection of mouse preimplantation embryos by *i*-GONAD were accompanied by surgery (exposure of ovary/oviducts/uterus) and operation/manipulation (DNA injection via oviductal wall and in vivo EP). All efforts were made to minimize the number of animals used and their suffering.

### 2.2. Preparation of Reagents Used for i-GONAD, si-GONAD, and simi-GONAD

For preparation of gRNAs targeting murine *GGTA1*, we designed the following two gRNAs: Ex4 gRNA (5′-GAGAAAATAATGAATGTCAA-3′) for targeting a portion (A site) spanning ATG in the exon 4 of *GGTA1* and #6 gRNA (5′-GAGAAAATAATGAATGTCAA-3′) for targeting a portion (B site) corresponding to the 3′ end terminal in the exon 4 of *GGTA1* (Figure 3A). Each CRISPR RNA (crRNA) for these gRNAs was synthesized by Integrated DNA Technologies, Inc. (IDT; Coralville, Iowa, USA) as Alt-R™ CRISPR crRNA product. For the preparation of gRNAs for targeted KI into murine *Mecp2*, the following two gRNAs were used: Mecp2-L2 (5′-CCCAAGGATACAGTATCCTA-3′) and Mecp2-R1 (5′-AGGAGTGAGGTCTAGTACTT-3′) [22]. For synthesis of donor ssODNs targeted to Mecp2, Mecp2-Left (Mecp2-L2-*lox*66) (5′-ccagcaacctaaagctgttaagaaatctttgggccccagcttgacccaaggatacagtatgctagcTACCGTTCGTATAATGTATGCTATACGAAGTTATcctagggaagttaccaaaatcagagatagtatgcagcagccaggggtctcatgtgtggca-3′) and Mecp2-Right (Mecp2-R1-*lox*71) (5′-ccactcctctgtactccctggcttttccacaatccttaaactgaaggagtgaggtctagtTACCGTTCGTATAGCATACATTATACGAAGTTATgaattcacttgggggtcattgggctagactgaata tctttggttggtacccagacctaatccacca-3′) were used as donors I and II, respectively [22]. These gRNAs and ssODNs were synthesized by IDT.

The crRNA (200 μM) was mixed with the trans-activating small RNA (tracrRNA) (200 μM; #99324036; purchased from IDT) in the ratio of 1:1. The mixture was subjected to annealing to generate tracr/crRNA complex (gRNAs).

For performing *i*-GONAD and *si*-GONAD using RNP targeted to *GGTA1*, 3 μL of Ex4 gRNA (or #6 gRNA) was incubated with 2 μL of Guide-it recombinant Cas9 protein (#1711676A; 3 μg/μL; TaKaRa Shuzo Co. Ltd., Shiga, Japan), 1 μL of fluorescein isothiocyanate-conjugated dextran (FITC-dextran), 3 kDa (hereafter referred to as FITC-dextran) (2 μg/μL; #D3306; Thermo Fisher Scientific, Waltham, MA, USA), and 1 μL of 0.4% Fast Green FCF (#15939-54; Nacalai tesque, Kyoto, Japan; used to enable visual identification of the injected site), in 3 μL of Opti-MEM (#31985062; Thermo Fisher Scientific).

As a control experiment, both gRNAs were simultaneously subjected to *i*-GONAD, which was thereafter called “simultaneous *i*-GONAD (*simi*-GONAD). For performing *simi*-GONAD using RNP targeted to *GGTA1*, 3 μL of Ex4 gRNA and 3 μL of #6 gRNA were incubated with 2 μL of Cas9 protein, 1 μL of FITC-dextran, and 1 μL of 0.4% Fast Green FCF. The final concentrations of gRNA and Cas9 protein in RNP were 30 μM and 0.6 μg/ μL, respectively.

For performing *si*-GONAD-mediated KI experiments, 3 μL of Mecp2-L2 gRNA (100 μM) was incubated with 2 μL Mecp2-Left donor I ssODN (100 μM), 2 μL of Cas9 protein, 1 μL of FITC-dextran (1 μg/μL), and 1 μL of 0.4% Fast Green FCF in 1 μL of Opti-MEM. For performing second *i*-GONAD, 3 μL of Mecp2-R1 gRNA (100 μM) was incubated with 2 μL Mecp2-Right donor II ssODN (100 μM), 2 μL of Cas9 protein, and 1 μL of 0.4% Fast Green FCF in 2 μL of Opti-MEM. For performing *simi*-GONAD-mediated KI experiments, a solution containing 3 μL of Mecp2-L2 gRNA (100 μM), 2 μL Mecp2-Left donor I ssODN (100 μM), 3 μL of Mecp2-R1 gRNA (100 μM), 2 μL Mecp2-Right donor II ssODN (100 μM), 1 μL of Cas9 protein, and 1 μL of 0.4% Fast Green FCF was prepared. In each solution, the final concentrations of gRNAs and ssODNs ranged from 25–30 μM and 17–20 μM, respectively.

### 2.3. si-GONAD, simi-GONAD, and i-GONAD

*i*-GONAD, *si*-GONAD, and *simi*-GONAD, which involve intraoviductal injection of a solution and subsequent in vivo EP, were performed under anesthesia after IP injecting the three combined anesthetics; medetomidine (0.75 mg/kg; Nippon Zenyaku Kogyo Co. Ltd., Fukushima, Japan), midazolam (4 mg/kg; Sandoz K.K., Tokyo, Japan), and butorphanol (5 mg/kg; Meiji Seika Pharma Co., Ltd., Tokyo, Japan).

The first *i*-GONAD was performed, following our protocols [24,25]. Briefly, the ovary/oviduct/uterus was exposed on the back of an anesthetized pregnant female (on Day 0.7 of pregnancy (approximately at 11:00 h)), followed by injection of a solution via the oviductal wall just in front of ampulla using a glass micropipette, which is connected to a mouthpiece. Then, the injected oviduct was covered with a small piece of wet paper (KimWipe; Jujo-Kimberly Co. Ltd., Tokyo, Japan) and immediately subjected to in vivo EP using a pair of disc-type tweezer-type electrodes (#CUY652-3; Nepa Gene Co., Ltd., Chiba, Japan), and the square-pulse generator NEPA21 (Nepa Gene Co., Ltd.) that generates poring pulse (Pp) and transfer pulse (Tp). The electric conditions are as follows: 3 Pp (50V/5 ms wavelength/50 ms duration/10 decay rate/± polarity) and 3 Tp (10V/50 ms wavelength/50 ms duration/40 decay rate/± polarity) that were previously determined by us [24,25]. After in vivo EP, the electroporated oviducts were returned to their original position and the abdominal wound was closed. The anesthetized mice were allowed to recover by subcutaneous injection of atipamezole (3.75 mg/kg; Nippon Zenyaku Kogyo Co. Ltd.), a medetomidine antagonist, and then warmed on an electric plate warmer. The detailed procedure involved in second *i*-GONAD (*si*-GONAD) is shown schematically in the right panel of Figure 1. On Day 1.7 of pregnancy (approximately 11:00 h), a new incision was made onto the dorsal skin of the female subjected to first *i*-GONAD near the site where the incision was made the previous day. Then, the ovary/oviduct/uterus was pulled out through the opened muscle layer and placed onto the paper towel. A solution (1–1.5 μL) containing RNPs was injected via the oviductal wall. After injection, in vivo EP was performed on the injected oviduct using the same conditions as those used for the first *i*-GONAD. After EP, the ovary/oviduct/uterus was returned to the original position and the incision was closed using suture wound clips.

For performing *si*-GONAD using fluorescent dextrans, a solution (1–1.5 μL) containing 1 μL of tetramethylrhodamine-dextran 3 kDa (hereafter referred to as rhodamine-dextran) (2 μg/μL; #D3307; Thermo Fisher Scientific) and 1 μL of 0.4% Fast Green FCF in 8 μL of Opti-MEM was used for the first *i*-GONAD. Then, the same female was subjected to the second *i*-GONAD using a solution containing 1 μL of FITC-dextran (1 μg/μL) and 1 μL of 0.4% Fast Green FCF, in 8 μL of Opti-MEM the next day. In some cases, the mice were subjected to second *i*-GONAD first, and then to the first *i*-GONAD. For performing *simi*-GONAD, a solution containing 1 μL of rhodamine-dextran (1 μg/μL), 1 μL of FITC-dextran (1 μg/μL), and 1 μL of 0.4% Fast Green FCF in 7 μL of Opti-MEM was used.

In the case of *si*-GONAD targeted to mouse *GGTA1*, a solution containing Cas9 and #6 gRNA was used for the first *i*-GONAD. Then, the same female was subjected to second *i*-GONAD using a solution containing Cas9 and Ex4 gRNA the next day.

In the case of the *si*-GONAD used for KI experiment, a solution containing Cas9, Mecp2-L2 gRNA, and Mecp2-Left donor I ssODN was used for the first *i*-GONAD. Then, the same female was subjected to second *i*-GONAD using a solution containing Cas9, Mecp2-R1 gRNA, and Mecp2-Right donor II ssODN.

### 2.4. Embryo Collection, ZP Removal, and Staining with Fluorescence-Labeled Lectin

The morula stage embryos (corresponding to 8-cell to 16-cell embryos) were collected from the oviducts of mice subjected to *i*-GONAD, *si*-GONAD, or *simi*-GONAD by flushing the entire oviducts with Dulbecco’s modified Ca^2+^, Mg^2+^-free phosphate-buffered saline (PBS) containing 0.3% fetal bovine serum (FBS) (hereafter referred to as PBS-FBS). The collected embryos were then subjected to fluorescence analysis under a fluorescence microscope or to ZP removal. For ZP removal, the ZP-intact embryos were immersed in an acid Tyrode’s solution (#T1788; Sigma-Aldrich, St. Louis, MO, USA) for a short period (approximately 30 s) at room temperature (~25 °C) prior to the mechanical removal of the ZP that is thinned. The ZP-free embryos were next subjected to staining with Alexa Fluor 594 (AF594)-labeled isolectin BS-I-B_4_ lectin (hereafter referred to as AF594-IB4) (1 μg/μL) for 15 min at room temperature, following our method [26] with a slight modification. After staining, the embryos were washed with PBS-FBS prior to observation under a fluorescence microscope.

### 2.5. Fluorescence Imaging

The freshly isolated morulae, and morulae stained with AF594-IB4, were directly subjected to fluorescence analysis under a fluorescence microscope (BX60; Olympus, Tokyo, Japan) using DM505 (BP460-490 and BA510IF; Olympus) and DM600 filters (BP545-580 and BA6101F; Olympus), which were used for the detection of FITC-derived green fluorescence and AF594-derived red fluorescence, respectively. The images were captured using a digital camera (FUJIX HC-300/OL; Fuji Film, Tokyo, Japan). The images were then printed using a digital color printer (CP700DSA; Mitsubishi, Tokyo, Japan).

### 2.6. Single Embryo Analysis

The morula stage embryos were individually subjected to genomic DNA isolation by placing an embryo into 1–2 μL of PBS-FBS in a 1.5-mL Eppendorf tube with the help of a mouthpiece-controlled micropipette as previously described [27,28]. The embryo was incubated with 15 μL of lysis buffer (0.125 μg/mL of proteinase K, 0.125 μg/mL of Pronase E, 0.32 M sucrose, 10 mM Tris-HCl (pH 7.5), 5 mM MgCl_2_, and 1% (v/v) Triton X-100) for 2 days at 37 °C. The genomic DNA of the embryos was extracted using the phenol/chloroform extraction method. The aqueous supernatant was subjected to ethanol precipitation. The precipitated DNA was dissolved in 6 μL of sterile water and stored at 4 °C.

The DNA sample was subjected to whole genome amplification (WGA) using the illustra GenomiPhi V2 DNA Amplification Kit (#25-6600-31; GE Health Care Japan, Tokyo, Japan) as previously described [27,28]. Briefly, 2 μL of genomic DNA was incubated with 8 μL reaction buffer containing enzyme in a 20-μL reaction volume overnight at 30 °C. The resulting WGA products (2 μL) were subjected to the first polymerase chain reaction (PCR).

To examine the presence of mutations in the target murine *GGTA1* gene, the following primers were used: mEx4-S (5′-GCAAATGTGGATGCTGGGAAC-3′; sense primer)/mEx4-RV (5′-ACAGTTTTAATGGCCATCTGG-3′; reverse primer) and nested primer set mEx4-2S (5′-TGAATCGAGCAGGTGTTTCAT-3′; sense primer)/mEx4-2RV (5′-AGGAACACAGGAAGACTGGAC-3′; reverse primer). The expected sizes of PCR products for the first PCR and nested PCR were 390 bp and 344 bp, respectively.

For checking KI event in the target murine *Mecp2* gene, the following primers were used: for the first PCR, Mecp2-F (5′-AAGAAGCCAACCATACAGTGC-3′; sense primer) and Mecp2-R (5′-GCTTGCTCAGAAGCCAAAAC-3′; reverse primer); for nested PCR, Mecp2-F2 (5′-CCAACCATACAGTGCTTACAT-3′; sense primer) and Mecp2-R2 (5′-TCA GAAGCCAAAA CAGCTGG-3′; reverse primer). The expected sizes of PCR products for the first PCR and nested PCR were 979 bp and 963 bp, respectively.

PCR was performed in a 20-μL reaction volume. The PCR conditions were as follows: initial denaturation (92 °C for 10 min), followed by 40 cycles of denaturation (96 °C for 10 s), annealing (56 °C for 1 min), and polymerization (72 °C for 2 min), and a final extension at 72 °C for 5 min using r*Taq* DNA polymerase (TaKaRa Taq; #R001A, Takara Shuzo). The PCR products (2 μL) were resolved on a 2% agarose gel and visualized by staining with ethidium bromide.

Direct sequencing of the PCR products was performed using a custom DNA sequencing service (Eurofins Genomics K.K., Tokyo, Japan).

### 2.7. Assay for the Floxed Allele

To detect the mutated *lox* (*lox66* and *lox71*) insertion in morulae, we employed the method previously described by Horii et al. [22], who performed PCR using Mecp2-F/Mecp2-R primers for the first PCR and Mecp2-2F/Mecp2-2R primers for nested PCR, which flank the target region (Figure 4A). The nested PCR products (approximately 960 bp in size) were digested with *Nhe* I and *Eco* RI, which cleave the floxed alleles. The enzyme-digested products were then subjected to electrophoresis using 2% agarose gels and visualized by staining with ethidium bromide.

## 3. Results

### 3.1. Experiment 1: si-GONAD Using Two Fluorescent Dextrans

The embryos subjected to *si*-GONAD must exhibit modifications in at least two sites in a target gene. Thus, the embryo must be transfected twice with the genome-editing components. To assess whether this event occurs after *si*-GONAD, we used two fluorescent (FITC- or rhodamine-labeled) dextrans to determine the numbers of zygotes that exhibit both FITC- and rhodamine-derived fluorescence when they are subjected to *si*-GONAD. For performing *si*-GONAD, rhodamine-dextran and Fast Green FCF (for monitoring successful intraoviductal injection) were first injected into the oviduct of a pregnant female at Day 0.7 of pregnancy, followed by in vivo EP. One day later, the same female was injected with FITC-dextran and Fast Green FCF, followed by in vivo EP. Also, *si*-GONAD was performed vice versa. As a control experiment, both dextrans were simultaneously subjected to *i*-GONAD, which was called *simi*-GONAD. As another control experiment, *i*-GONAD was performed on Day 0.7 females using FITC-dextran or rhodamine-dextran.

Two days (for *simi*-GONAD and *i*-GONAD) or one day (for *si*-GONAD) later, the morulae (at 8- to 16-cell stages) were isolated from the mice by flushing the oviducts with PBS-FBS. The morulae were subjected to fluorescence analysis. *i*-GONAD performed using only FITC-dextran yielded bright fluorescence in 61%–100% (11/18-12/12) of the embryos examined, although the fluorescence intensity varied among the embryos (arrows vs. arrowheads in Figure 2A(a–c)). Similar findings were also obtained in the embryos subjected to *i*-GONAD using only rhodamine-dextran. There were embryos showing rhodamine-derived fluorescence ranging from 65% (13/20) to 94% (15/16). Furthermore, the fluorescence intensity varied among the embryos (arrows vs. arrowheads in Figure 2A(d–f)). No fluorescence was detected in the intact embryos (Figure 2A(g–i)). The embryos subjected to *simi*-GONAD were normal, exhibiting both FITC- and rhodamine-derived fluorescence with efficiencies from 56% (5/9) to 64% (7/11) (arrows in Figure 2A(j–l)). The other embryos were those failing to show both fluorescence. When the intensity of fluorescence in each embryo was evaluated, there was 100% (20/20 examined) synchronicity (Figure 2B). In the group subjected to *simi*-GONAD, the embryos tended to incorporate both fluorescent molecules with an average efficiency of 60%. Contrastingly, the embryos subjected to *si*-GONAD exhibited both fluorescence (arrows in Figure 2A(m–o); Figure 2B) with 50% (10/20 examined) synchronicity, but 35% (7/20) of embryos exhibited either fluorescence (arrowheads in Figure 2A(m–o); Figure 2B). Thus, even in the groups subjected to *si*-GONAD with a 1-day interval between two *i*-GONAD steps, the embryos tended to incorporate both fluorescent molecules with an average efficiency of 50%. This suggests that indels can be introduced into at least two different sites in a target gene when *si*-GONAD is employed.

### 3.2. Experiment 2: Induction of Indels in the Two Sites (Which Are Located *Close to Each Other*) of Mouse GGTA1 (exon 4) by si-GONAD

In this experiment, we intended to disrupt two sites (called A and B) on the exon 4 of endogenous mouse *GGTA1* using *si*-GONAD. These A and B sites are recognized by Ex4 gRNA and #6 gRNA, respectively, and are located 44 bp apart from each other (Figure 3A). As mentioned previously, *GGTA1* encodes α-GalT, capable of synthesizing α-Gal epitope, a cell-surface carbohydrate. α-Gal epitope expression can be detected by staining the embryos with AF594-IB4 [29,30]. Thus, the embryos with completely disrupted *GGTA1* should be negatively stained with AF594-IB4. Previously, we had reported that the site recognized by Ex4 sgRNA in *GGTA1* can be successfully edited by *i*-GONAD with an efficiency of approximately 60% [23]. Thus, in this study, we subjected Day 0.7 B6C3F1 females to *i*-GONAD using a solution containing Cas9, #6 gRNA, FITC-dextran (for monitoring successful gene delivery), and Fast Green FCF as a control. In total, 11 morulae were successfully collected from one oviduct (#1R) of a pregnant female. Of these, 10 (91%) embryos exhibited normal morphology (Table 1). Most morulae (82%; 9/11) exhibited FITC-derived green fluorescence (arrowheads in Figure 3C(a,b); Appendix A). AF594-IB4 staining revealed that there were no IB4-negative embryos among the FITC-positive embryos (arrowheads in Figure 3C(b,c); Appendix A). The embryo was subjected to genomic DNA isolation and subsequently WGA. Further, the sequence spanning the B site was amplified by PCR and the PCR products were subjected to direct sequencing. The sequencing analysis revealed that 80% (8/10) of the embryos had indels in the B site (Table 1; Appendix A). These results suggest no strong correlation between α-Gal epitope expression and disrupted *GGTA1*, as long as α-Gal epitope expression was monitored at the morula stage.

Next, we performed *si*-GONAD to target the A and B sites on exon 4 of mouse *GGTA1* (Figure 3A). A solution containing Cas9, #6 gRNA, FITC-dextran, and Fast Green FCF was injected into the oviducts of B6C3F1 pregnant females. One day later, a solution containing Cas9, Ex4 gRNA, and Fast Green FCF was injected into the same oviducts. The morulae were isolated from 3 females subjected to *si*-GONAD on the next morning (Figure 3A). In total, 30 morulae were isolated from the females. Of these, 28 (93%) embryos exhibited normal morphology (Table 1). Among the 28 embryos, 89% (25/28) of morulae exhibited FITC-derived green fluorescence (arrowheads in Figure 3C(d,e); Appendix A). AF594-IB4 staining revealed that there were no IB4-negative embryos among the FITC-positive embryos (arrowheads in Figure 3C(e,f); Appendix A). After imaging, a single embryo was transferred to a 1-μL drop of PBS-FBS in a 1.5-mL tube for molecular biological analyses, such as PCR and direct sequencing (Figure 3B). Analysis of a single embryo revealed that 4% (1/28) of morulae had indels at the A site, but not at the B site, while 36% (10/28) of morulae had indels only at the B site (Table 1; Appendix A). Furthermore, 18% (5/28) of morulae had indels at both A and B sites, while 25% (7/28) of morulae had intact A and B sites (Table 1; Appendix A). The typical ideograms are shown in Figure 3D.

We also performed *simi*-GONAD to target the exon 4 of mouse *GGTA1*. A solution containing Cas9, Ex4 gRNA, #6 gRNA, FITC-dextran, and Fast Green FCF was injected into the oviducts of B6C3F1 pregnant females on Day 0.7 of pregnancy. Two days later, the morulae were isolated from three females subjected to *simi*-GONAD. In total, 24 morulae were obtained from three females. Of the 24 morulae, 92% embryos exhibited normal morphology (Table 1). Furthermore, 95% (21/22) of morulae exhibited FITC-derived green fluorescence (arrowheads in Figure 3C(g,h); Appendix A). AF594-IB4 staining revealed that there were no IB4-negative embryos among the FITC-positive embryos (arrowheads in Figure 3C(h,i); Appendix A). The sequence analysis of a single embryo demonstrated that none (0/22) of the morulae had indels at the A site, while 9% (2/22) of morulae had indels only at the B site (Table 1; Appendix A). None of the morulae had indels at both A and B sites. Surprisingly, most morulae (73%, 16/22) exhibited fluorescence and remained intact (Table 1; Appendix A).

### 3.3. Experiment 3: si-GONAD-Mediated KI of Mutated lox Sites into the Intronic Sequences Interposing Exon 3 of Mouse Mecp2 Gene

For testing the usefulness of *si*-GONAD-mediated KI of mutated *lox* sites in the target locus (*Mecp2*), we used all the genome-editing components (gRNAs and donor ssODNs) described by Horii et al. [22], who demonstrated that the sequential introduction of genome-editing components was useful for KI of mutated *lox* sites into the introns of *Mecp2* through in vitro EP (Figure 4A). In this study, *simi*-GONAD was also performed to evaluate if there was a difference in the KI ability between the two approaches.

For *si*-GONAD-mediated KI, a solution containing Cas9, Mecp2-L2 gRNA (targeted to the left intron), Mecp2-Left donor I ssODN (including *lox66* in which *Nhe* I site exists) for the intron 2, FITC-dextran and Fast Green FCF was first introduced into the oviducts of B6C3F1 pregnant females. One day later, a solution containing Cas9, Mecp2-R1 gRNA (targeted to the right intron), Mecp2-Right donor II ssODN (including *lox71* in which *Eco* RI site exists) for the intron 3 and Fast Green FCF was introduced into the same oviducts. For *simi*-GONAD, a solution containing Cas9, Mecp2-L2 gRNA, Mecp2-R1 gRNA, Mecp2-Left donor I ssODN, Mecp2-Right donor II ssODN, and Fast Green FCF was introduced into the oviducts of B6C3F1 pregnant females at Day 0.7 of pregnancy. After the surgery, the morulae were isolated and subjected to fluorescence analysis. Of the three females subjected to *si*-GONAD, 24 morulae were obtained from two females. Of the four females subjected to *simi*-GONAD, 31 morulae were obtained from two females (Table 2). In each case, >90% of embryos collected exhibited normal morphology (Table 2). More than 80% of morulae subjected to *si*-GONAD exhibited FITC-derived fluorescence (data not shown). Next, the genomic DNA isolated from the single embryo was subjected to PCR analysis using the Mecp2-F2 and -R2 primers (Figure 4A; see Materials and Methods). The PCR analysis revealed a band with a size of approximately 960 bp containing the sites recognized by the Mecp2-L2 and Mecp2-R1 gRNAs. In the group subjected to *si*-GONAD, the PCR analysis of the genomic DNA of 95% (21/22) of single embryos revealed a band with a size of approximately 960 bp (upper panel of Figure 4B). This suggested a very low frequency of LD of a sequence between the two sites sequences recognized by the Mecp2-L2 and Mecp2-R1 gRNAs. In the group subjected to *simi*-GONAD, the PCR analysis of the genomic DNA of 86% (24/28) of single embryos revealed a band with a size of approximately 300 bp (upper panel of Figure 4B; Table 2), which suggested a high frequency of LD in these samples. Direct sequencing of the 300-bp bands (*simi*-GONAD-KI-#3L-1) exhibiting LD demonstrated that the region surrounded by the two gRNA-recognizing sites had been entirely deleted (upper panel of Figure 4C).

We demonstrated that if KI into both target sites of *Mecp2* occurs successfully, digestion of 960-bp PCR products with *Nhe* I and *Eco* RI would generate approximately 170 and 160 bp fragments along with 630 bp fragment [22]. This is because the donor I and II contain the *Nhe* I and *Eco* RI sites, respectively (Figure 4A). If KI occurs in either of the target sites of *Mecp2,* only 170- or 160-bp band (but not the 630-bp band) would be generated after digestion with *Nhe* I and *Eco* RI. When the 960-bp bands obtained from the embryo subjected to *si*-GONAD and *simi*-GONAD were digested with those two enzymes, some samples (as exemplified by *si*-GONAD-KI-#1R-1 to -3 and -#1L-1 and -3) had the bands with a size ranging from 100 to 200 bp but no band corresponding to the size of 630-bp band was observed throughout the samples (lower panel of Figure 4B).

Direct sequencing of the 960-bp bands detected in most of the samples subjected to *si*-GONAD or *simi*-GONAD confirmed successful KI in one target site (recognized by Mecp2-L2 gRNA) of *Mecp2* in only one sample (*si*-GONAD-KI-#1L-1) (lower panel of Figure 4C; Table 2; Appendix A). The other samples were those with indels in either one or both target sites, or those with both intact sites (Table 2; Appendix A).

## 4. Discussion

*si*-GONAD requires two in vivo EP steps with a one-day interval between them, which may cause deleterious effects on the embryo survival. However, this study demonstrated that there were no marked damages in the embryos, as most of the embryos collected from the oviducts subjected to *si*-GONAD exhibited normal morphology (91%–93% for Experiment 2 and 78%–100% for Experiment 3). Thus, we concluded that *si*-GONAD is not harmful to the development of early embryos, at least from zygotes to the morula stage in the B6C3F1 female mice.

It seems to be difficult to introduce two independent indels at the two target sites that are located very close to each other (44 bp apart) through one-step gene delivery of CRISPR/Cas9 components. This is because the two types of RNPs introduced within an embryo would compete with each other for the target sites. Only a small proportion of morulae (9%) subjected to *simi*-GONAD exhibited mutations at the B sites (recognized by #6 gRNA) in the mouse *GGTA1* and the A site (recognized by Ex4 gRNA) was unaltered (see Table 1; Appendix A). Furthermore, indels were induced in both A and B sites of the morulae subjected to *simi*-GONAD (see Table 1; Appendix A). The induction of indels in both A and B sites of morulae subjected to *si*-GONAD was evident with an efficiency of 18% (see Table 1; Appendix A). Particularly, 36% of the morulae recovered from the #2R oviduct exhibited indels in both A and B sites (see Table 1; Appendix A). Notably, 36% of embryos exhibited indels only in the B site (see Table 1), which indicated that indel induction might have occurred after first (but not second) *si*-GONAD. Similarly, 4% of embryos exhibited indels only in the A site (see Table 1), which indicated that indels might have occurred during the second (but not first) *si*-GONAD. In total, 40% of the morulae subjected to *si*-GONAD had indels either at A or B site on exon 4 of *GGTA1*. This suggested that *si*-GONAD, which involves two in vivo EP steps, may be more preferable for increasing the probability of inducing genome-editing at the target site than *i*-GONAD, which involves a single in vivo EP step.

In Experiment 2, we monitored the expression of α-Gal epitope on the cell-surface of morulae that were subjected to CRISPR/Cas9-mediated α-GalT gene disruption. However, complete ablation of α-Gal epitope, which is determined by AF594-IB4 staining, was not observed in the morulae subjected to *si*-GONAD that had mutated in A site or B site or both (see arrowheads in Figure 3C(e,f)). This was an unpredictable finding because we had previously reported that porcine embryos (morulae to blastocysts), which were derived from oocytes cytoplasmically injected with Cas9 mRNA and gRNA targeting exon 4 of porcine *GGTA1* or those reconstituted via somatic cell nuclear transfer (SCNT) by genetically modified fibroblasts lacking α-Gal epitope expression, exhibited marked downregulation of α-Gal epitope expression on their surface [31,32]. α-Gal epitope is reported to be expressed on the ZP and cell-surface of unfertilized porcine oocytes [33], which suggests that α-Gal epitope is maternally inherited in pigs. Thus, it is conceivable that α-Gal epitope, which was detected in the *GGTA1*-ablated mouse morulae, may be maternally inherited.

In Experiment 3, we failed to generate morulae with KI alleles in which floxed sites had been knocked-in in both sides of the introns interposing exon 3 of *Mecp2* (see Table 2; Appendix A). Only a morula with one floxed site in the 5′ site of *Mecp2* was successfully generated (see lower panel of Figure 4C; Table 2; Appendix A). We used the gRNAs and donor ssODNs in our previous paper [22]. Notably, these genome-editing components generated embryos (blastocysts) with two floxed sites in the target *Mecp2* locus with an efficiency of 21% (33/155 tested) after in vitro EP (with seven electric pulses for both first and second EP) in the presence of 5 μL of drop containing Cas9/gRNA/ssODN (100/24/400 ng/μL) mixture. In this condition, the efficiency of generating LD was 36% (56/155 tested). Notably, EP performed with five electric pulses (for both first and second EP) resulted in the generation of blastocysts with two floxed sites with 8% (3/40 tested) efficiency, and the LD efficiency was 30% (12/40 tested). These findings suggest that the elevated number of electric pulses can increase the rate of generation of two floxed sites. We usually use five electric pulses for KI experiments and indel induction during *i*-GONAD in the NEPA21 apparatus [24,25]. Thus, the elevated number of electric pulses appears to be beneficial for the incorporation of large amounts of genetic material into an embryo, which also promotes KI. Additionally, the concentration of ssODN may be one of the important factors determining the success of KI, as suggested by us [24]. In this study, we used 20 μM (1000 ng/μL) of ssODN when *i*-GONAD-based KI was performed as recommended for KI induction in the *i*-GONAD system [24]. The use of drugs that are capable of promoting homology direct repair (HDR) or suppressing non-homologous end joining (NHEJ) would also be promising, as suggested by Maruyama et al. [34], who demonstrated that co-injection of murine zygotes with a mixture containing Cas9 mRNA, gRNA, template ssODNs, and Scr7 (an inhibitor for DNA ligase IV) significantly improved the efficiency of HDR-mediated KI.

## 5. Conclusions

In this study, we demonstrated that the introduction of two fluorescence-labeled dextrans into an embryo is possible through two intraoviductal instillations of fluorescent substances (each followed by in vivo EP) with a 1-day interval between the two instillations. Furthermore, *si*-GONAD enabled the induction of mutations (indels) in the two target sites (which are located very close to each other) in the exon 4 of *GGTA1*. Contrastingly, no such event was observed in the embryos subjected to *simi*-GONAD. We also attempted to perform KI of two mutated *lox* sites into the intronic portion flanking the exon 3 of *Mecp2* using *si*-GONAD. However, we failed to produce embryos with floxed alleles. Only one sample had one floxed site on the target area. None of the embryos subjected to *simi*-GONAD had the floxed KI sites. Although *si*-GONAD is required for further improvement for successful KI in both sides on the target area, it has greater potential than the previously described *i*-GONAD (which is based on a single delivery of CRISPR components). For example, it must be theoretically possible to perform ssODN-based KI via *si*-GONAD at the nearby site, where indels have already been induced. *si*-GONAD may also be useful for increasing the success rate of generation of indels, because when the first GONAD fails to induce indels, the second GONAD would compensate for the failure through successful indel induction in the target locus.

## Figures and Tables

**Figure 1 cells-09-00546-f001:**
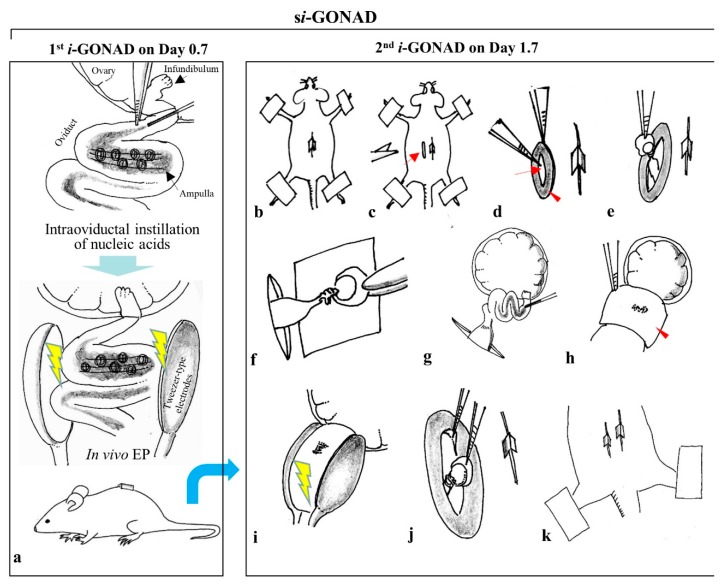
Schematic representation of the detailed procedure of sequential improved genome-editing via oviductal nucleic acid delivery (*si*-GONAD). The oviducts of a pregnant female were subjected to improved genome-editing via oviductal nucleic acid delivery (*i*-GONAD), as shown in the left panel (**a**) on Day 0.7 of pregnancy (approximately 11 h). The day when the vaginal plug is recognized is defined as Day 0 of pregnancy. Then, the same female is subjected to second *i*-GONAD, as shown in the right panel on Day 1.7 of pregnancy (~11:00 h). For performing second *i*-GONAD, a new incision (arrow in (**c**)) is made onto the dorsal skin near the site where the incision was made the previous day. Then, the portion of muscle layer (arrowhead in (**d**)), where the incised part remains open (without closure) after surgery, which is covered soon after surgery with hyaluronan-rich matrix, is opened by removing the sheets using forceps (arrow in (d)). Next, the ovary/oviduct/uterus is pulled out and placed onto the paper towel (**e**,**f**). A solution (1–1.5 μL) containing nucleic acids and First Green FCF is injected via the oviductal wall, which is located immediately above the ampulla (**g**). After injection, the injected site is covered with a small piece of wet KimWipe paper (arrow in (**h**)). The injected site is easily visible through the paper as blue spots. Then, the entire oviduct is held using tweezer-type electrodes and is subjected to in vivo EP (**i**). After EP, the ovary/oviduct/uterus is returned to the original position (**j**) and the incision is closed using suture wound clips (**k**). Abbreviations: *i*-GONAD, improved genome-editing via oviductal nucleic acids delivery; *si*-GONAD, sequential *i*-GONAD; EP, electroporation.

**Figure 2 cells-09-00546-f002:**
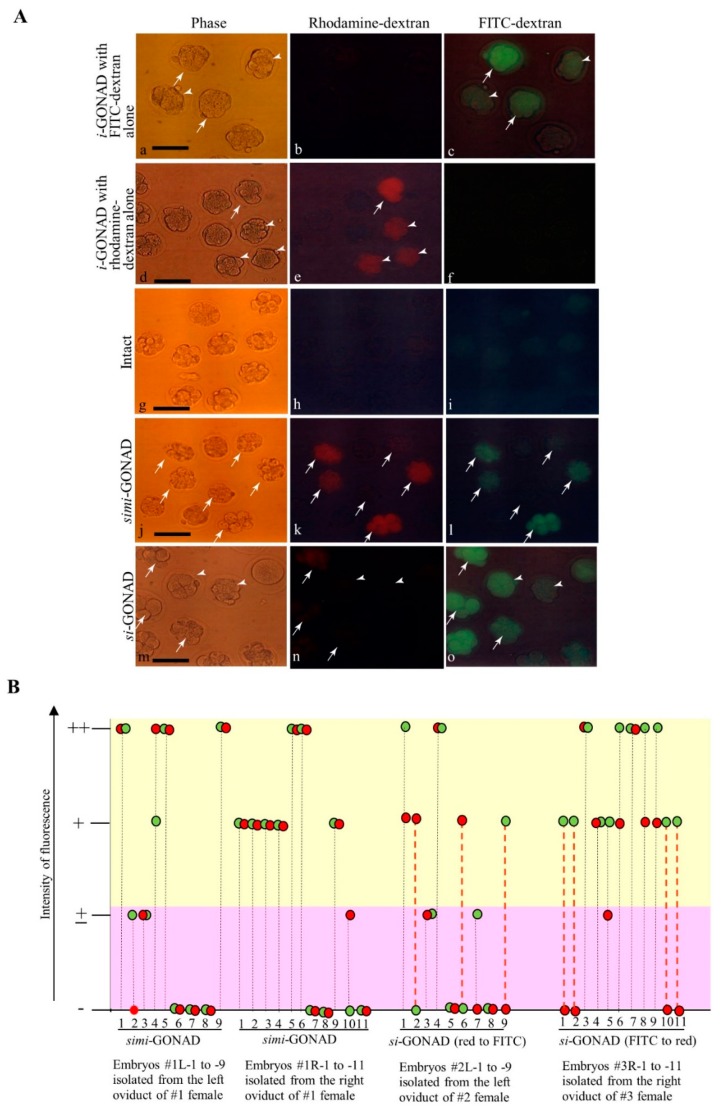
*si*-GONAD and *simi*-GONAD using two fluorescent dextrans. **A.** Fluorescence analysis of morulae was performed after *i*-GONAD with rhodamine-dextran alone (**d**–**f**), *i*-GONAD with FITC-dextran alone (**a**–**c**), *si*-GONAD in which oviducts are first instilled with rhodamine-dextran, and then with FITC-dextran (**m**–**o**), and *simi*-GONAD with both rhodamine-dextran and FITC-dextran (**j**–**l**). The untreated morulae were also subjected to fluorescence analysis (**g**–**i**). Phase-contrast images captured under light microscope. Images of rhodamine-dextran were captured under UV illumination to detect rhodamine-derived red fluorescence. Images of FITC-dextran were captured under UV illumination to detect FITC-derived green fluorescence. Scale bar = 100 μm. **B.** Summary of fluorescence intensity in each embryo shown in **A**. For *si*-GONAD (red to FITC), oviducts were first instilled with rhodamine-dextran prior to in vivo EP and one day later, the same oviducts were instilled with FITC-dextran. For *si*-GONAD (FITC to red), oviducts were first instilled with FITC-dextran prior to in vivo EP. One day later, the same oviducts were again instilled with rhodamine-dextran. For each group, more than nine morulae were collected from an oviduct and subjected to fluorescence analysis under a fluorescence microscope. The fluorescence intensity was evaluated as; (no fluorescence), + (faint fluorescence), + (moderate fluorescence), and ++ (strong fluorescence). Synchronicity was judged as YES (which is shown by the black dotted line) when an embryo synchronously exhibited both FITC and rhodamine fluorescence, as exemplified by the following relationship: ++ vs. ++, ++ vs. +, + vs. +, - vs. – or – vs. +. In contrast, it was judged as NO (which is shown by the red dotted line), when an embryo exhibited both FITC and rhodamine fluorescence with different degrees of intensity, as exemplified by the following relationship: ++ vs. -, ++ vs. + or + vs. -. Notably, the fluorescence intensity in each embryo was well synchronized when *simi*-GONAD is performed. However, the synchronicity varied when embryos were subjected to *si*-GONAD. Abbreviations: *i*-GONAD, improved genome-editing via oviductal nucleic acids delivery; *si*-GONAD, sequential *i*-GONAD; *simi*-GONAD, simultaneous *i*-GONAD; FITC, fluorescein isothiocyanate.

**Figure 3 cells-09-00546-f003:**
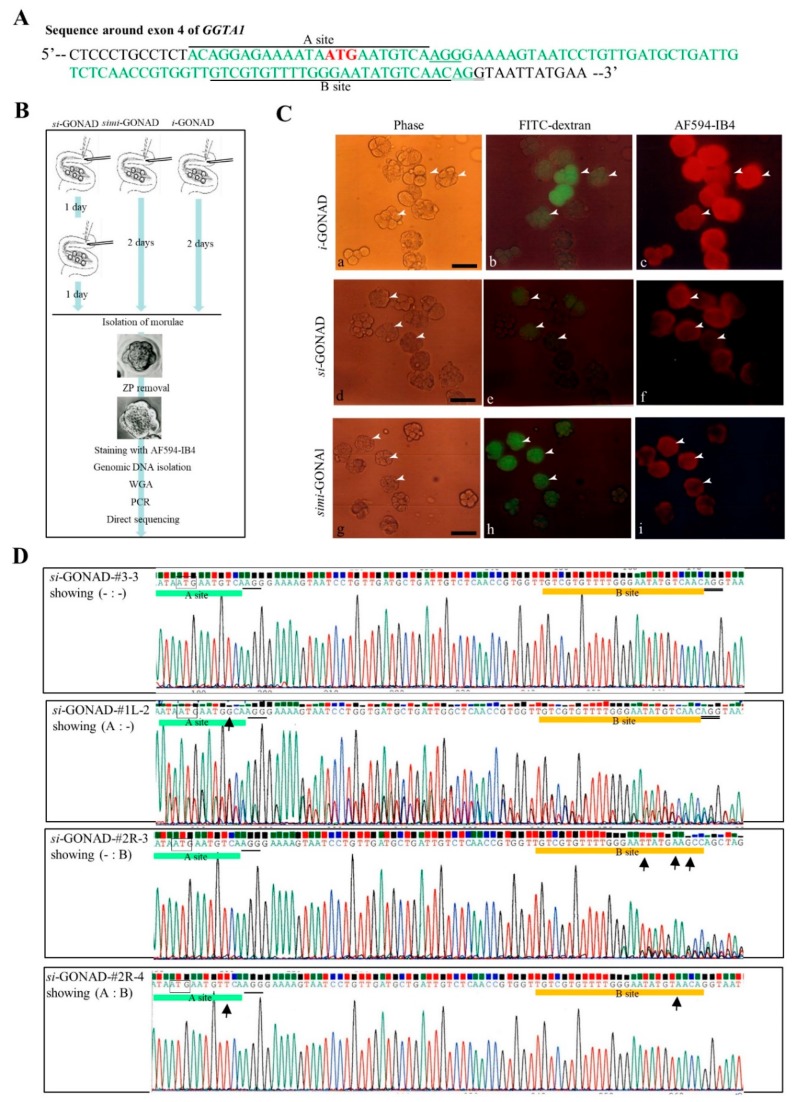
*si*-GONAD, simultaneous *i*-GONAD (*simi*-GONAD) and *i*-GONAD using Cas9 complexed with gRNAs targeted to murine *GGTA1*. **A.** A sequence around the exon 4 of *GGTA1*. A site (recognized by Ex4 gRNA) and B site (recognized by #6 gRNA) are indicated using underlines. Translation initiation site (ATG) is shown in red. Protospacer adjacent motif (PAM) sites are indicated by a single underline (for Ex4 gRNA) or by double underlines (for #6 gRNA). Sequence in the coding region of exon 4 in *GGTA1* is shown in green. **B.** Flowchart of the experiments used for testing the feasibility of *si*-GONAD for inducing two indels at the target sites, which are located very close to each other. **C.** Fluorescence analysis of morulae isolated from the females subjected to *i*-GONAD using a solution containing Cas9, #6 gRNA (**a–c**), and *si*-GONAD (**d–f**), followed by *i*-GONAD using a solution containing Cas9/Ex4 gRNA complex, or *simi*-GONAD using a solution containing Cas9, #6 gRNA, and Ex4 gRNA (**g–i**). The isolated morulae are first subjected to ZP removal, followed by staining with AF594-IB4. Notably, morulae exhibiting fluorescein isothiocyanate-conjugated (FITC)-derived green fluorescence are also stained with AF594-IB4 (see arrowheads in **a** to **i**). Phase-contrast images were captured under a light microscope. Images of FITC-dextran were captured under UV illumination to detect the FITC-derived green fluorescence. AF594-IB4 staining images were captured under UV illumination to detect the AF594-IB4–derived red fluorescence. Scale bar = 100 μm. **D.** Direct sequencing of PCR products of embryos subjected to *si*-GONAD using primers specific to the *GGTA1* gene. Examples showing intact (- : -), indels only at the A site (A : -), only at the B site (- : B), and at both sites (A : B). Arrows below ideograms indicate the sites with indels. The translation initiation codon, ATG, is shown within boxes. PAM is indicated with black underlines. The A and B sites are shown in green and orange lines, respectively. Abbreviations: *i*-GONAD, improved genome-editing via oviductal nucleic acids delivery; *si*-GONAD, sequential *i*-GONAD; *simi*-GONAD, simultaneous *i*-GONAD; ZP, zona pellucida; indels, insertion-deletion mutations; WGA, whole genome amplification; PCR, polymerase chain reaction.

**Figure 4 cells-09-00546-f004:**
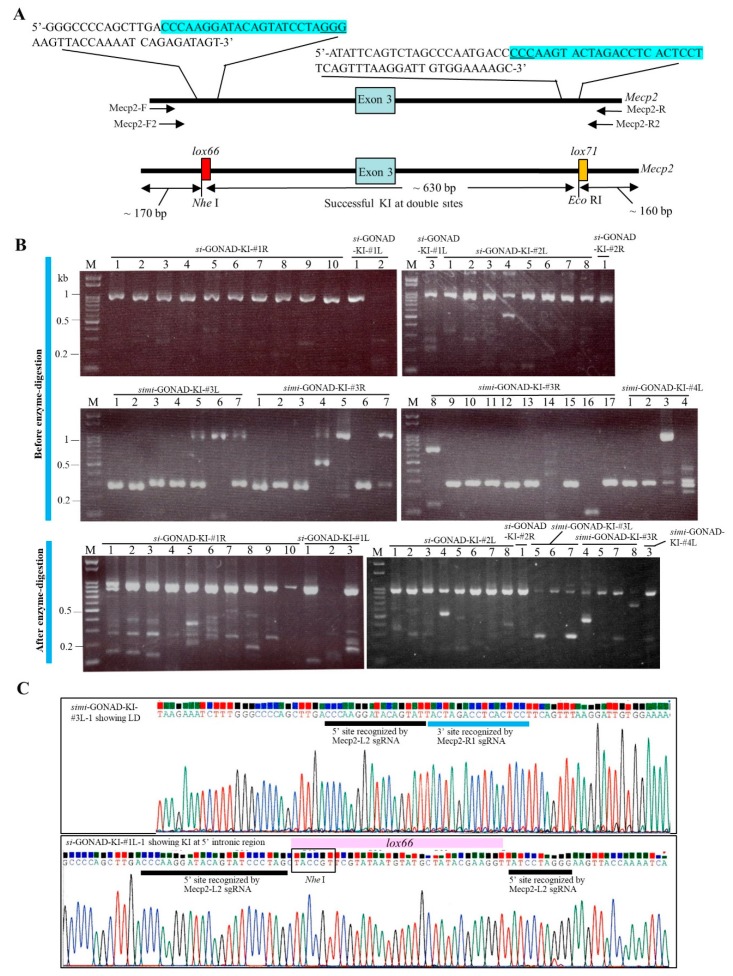
*si*-GONAD and *simi*-GONAD for KI of mutated *lox* sites into the intronic sequences flanking the exon 3 of murine *Mecp2*. **A**. The structure of intron 2, exon 3, and intron 4 of *Mecp2* (upper panel) and that of *Mecp2* with successful KI at double sites (lower panel). In the upper panel, the sequences near the sites recognized by Mecp2-L2 gRNA and Mecp2-R1 gRNA are shown in a blue shadow above the structure of *Mecp2*. Primers (Mecp2-F/Mecp2-R and Mecp2-F2/Mecp2-R2) used for PCR and nested PCR are shown below the structure of *Mecp2*. In the lower panel, restriction enzyme sites (*Nhe* I and *Eco* RI sites) are shown when successful KI is performed at both target sites. **B.** Electrophoretic pattern of PCR products derived from a single morula using primer sets specific to *Mecp2*. In the upper panel, undigested PCR products of samples subjected to *si*-GONAD and *simi*-GONAD are shown. Notably, a band of approximately 960-bp size is evident in the samples subjected to *si*-GONAD (which is denoted as “*si*-GONAD-KI”), while this band is scarce in the samples subjected to *si*-GONAD (which is denoted as “*simi*-GONAD-KI”). In the lower panel, PCR products (derived from both *si*-GONAD and *simi*-GONAD) with a size of 960-bp after digestion with *Nhe* I and *Eco* RI are shown. Notably, the cleaved products (170 to 160 bp), probably from the successfully knocked-in *lox* sites, are discernible in some samples (i.e., *si*-GONAD-KI-#1R-1, -2, -3 and -8, and *si*-GONAD-KI-#1L-3). Unfortunately, the 630-bp band, which suggests double KI at the *Mecp2* locus, was not observed in the samples tested. M, 100 bp-ladder markers. **C.** Direct sequencing of PCR products derived from the genomic DNA of females subjected to *si*-GONAD or *simi*-GONAD using primers specific to the *Mecp2* gene. *simi*-GONAD-KI-#3L-1 sample, in which LD was suggested based on the electrophoretic pattern (see the lane “simi-GONAD-KI-#3L-1” in the upper panel of Figure 4B), had a sequence lacking a region surrounded by the two sites (upper panel). The *si*-GONAD-KI-#1L-1 sample exhibited successful KI in one target site (recognized by Mecp2-L2 gRNA) of *Mecp2* (lower panel). Abbreviations: *i*-GONAD, improved genome-editing via oviductal nucleic acids delivery; *si*-GONAD, simultaneous *i*-GONAD; *simi*-GONAD, simultaneous *i*-GONAD; KI, knock-in.

**Table 1 cells-09-00546-t001:** Summary of *si*-GONAD, *simi*-GONAD, and *i*-GONAD targeted to exon 4 of murine *GGTA1*.

Method ^1^	Name of Oviducts Examined ^2^	Rate of Normal Morulae (%) ^3^	Total No. of Morulae Examined			Mode of Mutations ^4^			
				(A : -) (%)	(- : B) (%)	(A : B) (%)	LD (%)	Complex (%)	(- : -) (%)
*si*-GONAD	#1L	11/11 (100)	11	1 (9)	5 (45)	1 (9)	0 (0)	2 (18)	2 (18)
	#2R	11/13 (85)	11	0 (0)	5 (45)	4 (36)	1 (9)	1 (9)	0 (0)
	#3R	6/6 (100)	6	0 (0)	0 (0)	0 (0)	0 (0)	1 (17)	5 (83)
Total		28/30 (93)	28	1 (4)	10 (36)	5 (18)	1 (4)	4 (14)	7 (25)
*simi*-GONAD	#1L	10/12 (83)	10	0 (0)	0 (0)	0 (0)	0 (0)	2 (20)	8 (80)
	#1R	12/12 (100)	12	0 (0)	2 (17)	0 (0)	0 (0)	2 (17)	8 (67)
Total		22/24 (92)	22	0 (0)	2 (9)	0 (0)	0 (0)	4 (18)	16 (73)
*i*-GONAD	#1R	10/11 (91)	10	0 (0)	8 (80)	0 (0)	1 (10)	1 (10)	0 (0)

^1^*si*-GONAD was performed using a solution containing Cas9/#6 gRNA complex and FITC-dextran for first *i*-GONAD, and a solution containing Cas9/Ex4 gRNA complex for second *i*-GONAD. *simi*-GONAD was performed using a solution containing Cas9/#6 gRNA complex, FITC-dextran, and Cas9/Ex4 gRNA complex. *i*-GONAD was performed using a solution containing Cas9/#6 gRNA complex and FITC-dextran. ^2^ For each *si*-GONAD and *simi*-GONAD experiment, the morulae were collected from three plug-positive females (total five oviducts). For *i*-GONAD experiment, the morulae were collected from one plug-positive female. However, embryos were not obtained from some oviducts. ^3^ Rate of normal morulae (%) is determined as the number of embryos exhibiting normal morphology among the total number of embryos collected. ^4^ Mode of mutations (indels) is determined by direct sequencing of the PCR products derived from a single embryo. Mutations at the A or B site of exon 4 of *GGTA1* are defined as (A : -) and (- : B), respectively. Mutations induced at both sites are defined as (A : B). “Complex” indicates that several mutations are mixed in one embryo. LD indicates a large deletion in a region spanning exon 4 of *GGTA1*. Embryo exhibiting no mutation is defined as (- : -); Abbreviations: *i*-GONAD, improved genome-editing via oviductal nucleic acids delivery; *si*-GONAD, sequential *i*-GONAD; *simi*-GONAD, simultaneous *i*-GONAD; LD, large deletion.

**Table 2 cells-09-00546-t002:** Summary of *si*-GONAD and *simi*-GONAD-based KI targeted to intronic region flanking exon 3 of murine *Mecp2*.

Method ^1^	No. Oviducts Examined ^2^	Rate of Normal Morulae (%) ^3^	Total no. of Morulae Examined			Mode of KI ^5^			
				(KI : Indels) (%)	(Indels : -) (%)	(- : Indels) (%)	(Indels : Indels) (%)	LD (%)	(- : -) (%)
*si*-GONAD	#1R	10/11 (91)	10	0 (0)	1 (10)	2 (20)	1 (10)	0 (0)	6 (60)
	#1L	3/3 (100)	3 ^4^	1 (33)	0 (0)	0 (0)	1 (33)	1 (33)	0 (0)
	#2L	8/9 (89)	8	0 (0)	0 (0)	3 (38)	1 (13)	0 (0)	4 (50)
	#2R	1/1 (100)	1	0 (0)	1 (100)	0 (0)	0 (0)	0 (0)	0 (0)
Total		22/24 (92)	22	1 (5)	2 (9)	5 (23)	3 (14)	1 (5)	10 (45)
*simi*-GONAD	#3L	7/9 (78)	7	0 (0)	0 (0)	0 (0)	0 (0)	6 (86)	1 (14)
	#3R	17/18 (94)	17	0 (0)	2 (12)	0 (0)	1 (6)	14 (82)	0 (0)
	#4L	4/4 (100)	4	0 (0)	0 (0)	0 (0)	0 (0)	3 (75)	1 (25)
Total		28/31 (90)	28	0 (0)	2 (7)	0 (0)	1 (4)	23 (82)	1 (4)

^1^*si*-GONAD was performed using a solution containing Cas9, Mecp2-L2 gRNA, donor I ssODN, and FITC-dextran for first *i*-GONAD, and a solution containing Cas9, Mecp2-R1 gRNA, and donor II ssODN for second *i*-GONAD. *simi*-GONAD was performed using a solution containing Cas9, Mecp2-L2 gRNA, donor I ssODN, Mecp2-R1 gRNA, and donor II ssODN. ^2^ For each *si*-GONAD and *simi*-GONAD experiment, seven oviducts are were used to collect the morulae. ^3^ Rate of normal morulae (%) is determined as the number of embryos exhibiting normal morphology among the total number of embryos collected. ^4^ The 960-bp product was not detected in one sample (si-GONAD-KI-#1L-2; see the upper panel of Figure 4B), which is probably due to the large deletion of a sequence that is not recognized by the PCR primer set. ^5^ Mode of KI or mutations (indels) is determined by direct sequencing of the 960-bp PCR products derived from a single embryo. Samples with LD are determined by subjecting the PCR products to 2% gel electrophoresis. The samples lacking relatively large-sized region are included as those with indels. The samples showing complex patterns of ideograms suggest the presence of several mutations in one embryo. Thus, these samples are classified as “Indels.” KI on the 5′ intronic site and the indels on the 3′ intronic site of *Mecp2* are defined as (KI : Indels). Indels on either 5′ or 3′ site of *Mecp2* are defined as (Indels : -) or (- : Indels), respectively. Indels induced on both sites are defined as (Indels : Indels). Embryo exhibiting no mutation is defined as (- : -); Abbreviations: *i*-GONAD, improved genome-editing via oviductal nucleic acids delivery; *si*-GONAD, sequential *i*-GONAD; *simi*-GONAD, simultaneous *i*-GONAD; LD, large deletion; indels, insertion-deletion mutations; KI, knock-in.

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
