# Peer review of "Sequential i-GONAD: An Improved In Vivo Technique for CRISPR/Cas9-Based Genetic Manipulations in Mice"

_cells, 2020, doi:10.3390/cells9030546_

Round 1

Reviewer 1 Report

Sato and colleagues describe sequential i-GONAD as a novel in vivo Technique for CRISPR/Cas- mediated  genetic manipulations in mice. i-GONAD is based on intraoviductal instillation of a solution containing genome-editing components via a glass micropipette followed by in vivo electroporation. The novel sequential i-GONAD (si-GONAD) system described here uses electroporation on two different days, which allows more complex genetic manipulation e.g. introduction of 2 LoxP sites. In contrast to classical in vitro transgenic approaches, the sequential i-GONAD procedure allows to work in situ, which clearly reduces time and workload.

The manuscript is extremely well and carefully written. All experiments are presented in great detail. The only challenge for future experiments remains the successful introduction of the 2 proposed LoxP sites, which failed in the present manuscript. Nevertheless, all other reported experiments worked very well and are convincingly presented.

I found only the mistake that the arrows in Figure 3C are not explained.

Overall, I wish only to congratulate the authors to the careful study and manuscript.

Author Response

Comments and Suggestions for Authors

Sato and colleagues describe sequential i-GONAD as a novel in vivo Technique for CRISPR/Cas-mediated genetic manipulations in mice. i-GONAD is based on intraoviductal instillation of a solution containing genome-editing components via a glass micropipette followed by in vivo electroporation. The novel sequential i-GONAD (si-GONAD) system described here uses electroporation on two different days, which allows more complex genetic manipulation e.g. introduction of 2 LoxP sites. In contrast to classical in vitro transgenic approaches, the sequential i-GONAD procedure allows to work in situ, which clearly reduces time and workload. The manuscript is extremely well and carefully written. All experiments are presented in great detail. The only challenge for future experiments remains the successful introduction of the 2 proposed LoxP sites, which failed in the present manuscript. Nevertheless, all other reported experiments worked very well and are convincingly presented. I found only the mistake that the arrows in Figure 3C are not explained. Overall, I wish only to congratulate the authors to the careful study and manuscript.

Answer: We are thankful for the reviewer’s cordial comments that will encourage us to proceed with our on-going GONAD-related experiments.

As indicated by the reviewer, we have now included the following sentence —“Notably, morulae exhibiting FITC-derived green fluorescence are also stained with AF594-IB4 (see arrowheads in Fig. 3C-a-i)”— in the legend of Figure 3C of the revised text.

Reviewer 2 Report

Manuscript of Sato et.al. have improved the i-GONAD technology in terms of inducing two types of indels and two target sites that are located very close to each other via two subsequent in vivo electroporation at Day 0.7 of pregnancy and the day after (at Day 1.7 of pregnancy). The manuscript has written at a high methodological level and can be the starting point for developing an approach for introducing loxp sites in vivo.

I have minor suggestions/revisions.

  1. Why did authors use 3 kDa molecules (FITC-dextran or rhodamine-dextran) as the controls to measure electroporation efficiency since these molecules are smaller in size than Cas9 protein or RNP (~162 kDa). Thus both of dextrans could easier than Cas9 penetrate through the embryo membrane?
  2. Authors obtained maximal efficiency of i-GONAD (61-100%) with FITC dextran and 55% efficiency after si-GONAD. How these results could be translated in the efficiency with RNP electroporation?
  3. Why didn’t the authors use the equivalent for Cas9 protein (for example EGFP protein, 3 kDa)?
  4. Does any dextans or Cas9 penetrate («Cas 9 leakage») to the oviducts itself via i-GONAD/simi-GONAD/si-GONAD? Possibly it needs to be provided by immunofluorescence analysis. 
  5. Does and how the “Cas 9 leakage” affect the development of the embryo of the maternal organism, also changed by in utero electroporation?
  6. Could authors provide the data of intact genomic loci where theoretically could be observed off-target Cas9 activity besides the overall normal embryo morphology?
  7. Could the number of nonspecific modification rise since the time/concentration dependence of Cas9-RNP occurrence in the embryo through the development stages?
  8. How many days does the Cas9 exist in embryo? Is it possible to measure the titer of Cas9 after 0.7 to 1.7 day transition? 
  9. Which characteristics of embryo were selected to provide it integrity?
  10. Will the development of such an embryo be investigated further?

Author Response

Comments and Suggestions for Authors

Manuscript of Sato et.al. have improved the i-GONAD technology in terms of inducing two types of indels and two target sites that are located very close to each other via two subsequent in vivo electroporation at Day 0.7 of pregnancy and the day after (at Day 1.7 of pregnancy). The manuscript has written at a high methodological level and can be the starting point for developing an approach for introducing loxp sites in vivo.

I have minor suggestions/revisions.

  1. Why did authors use 3 kDa molecules (FITC-dextran or rhodamine-dextran) as the controls to measure electroporation efficiency since these molecules are smaller in size than Cas9 protein or RNP (~162 kDa). Thus both of dextrans could easier than Cas9 penetrate through the embryo membrane?

Answer 1: We employed rhodamine-dextran 3 kDa to monitor successful i-GONAD based on the report by Kaneko et al. (Sci Rep 4, 6382 (2014) https://doi.org/10.1038/srep06382) who used these molecules for the development of in vitro electroporation-mediated production of genome-edited rats. We have already mentioned the usefulness of rhodamine-dextran 3 kDa in our previous paper by Takabayashi et al. (Sci Rep 8: 12059, 2018. DOI:10.1038/s41598-018-30137-x), in which the following sentence—“Rhodamine is used to evaluate gene delivery in rat embryos upon in vitro electroporation9. “—is described.

As pointed out by the reviewer, the transfer of rhodamine-dextran 3 kDa molecule to embryos through zona pellucida and cell membrane is easier than that of larger- sized substances such as Cas9. In our preliminary test, rhodamine-dextran (70 kDa)(#D1818; Thermo Fisher Scientific) is found to be easily transferred to zona pellucida-intact zygotes after i-GONAD. Molecules with molecular size up to ~162 kDa may be taken up by zygotes after i-GONAD, but it seems difficult to introduce larger-sized molecules such as 5-kb plasmids (3,000 kDa).

  1. Authors obtained maximal efficiency of i-GONAD (61-100%) with FITC dextran and 55% efficiency after si-GONAD. How these results could be translated in the efficiency with RNP electroporation?

Answer 2: Generally, i-GONAD-mediated delivery of fluorescent substances to murine zygotes varies among pregnant females. For instance, in one mouse the efficiency was 61%, whereas in another the efficiency was 100%. On introduction of two fluorescence-labelled dextrans into zygotes using simi-GONAD, the % of embryos having both dextrans varied from 56% (5/9) to 64% (7/11). Similarly, the % of embryos having both dextrans varied from 33% (3/9) to 64% (7/11), when si-GONAD was performed. Thus, we consider that through simi-GONAD and si-GONAD, 50-60% of embryos might have successfully received both dextrans. These rates may also be true for the case of introduction of RNP into zygotes. In relation to this point, we rewrote the preceding sentence (see L314-327 in the revised text).

  1. Why didn’t the authors use the equivalent for Cas9 protein (for example EGFP protein, 3 kDa)?

Answer 3: The use of EGFP protein as an alternative to Cas9 to test the feasibility of our GONAD system is a fitting suggestion. However, we were not aware of its commercial availability at least at the time we started to develop GONAD in 2014.

  1. Does any dextans or Cas9 penetrate («Cas 9 leakage») to the oviducts itself via i-GONAD/simi-GONAD/si-GONAD? Possibly it needs to be provided by immunofluorescence analysis.

Answer 4: When we perform i-GONAD/si-GONAD/simi-GONAD, blue dye such as Trypan Blue or Fast Green is always included in the solution before injecting it into the lumen of an oviduct. The presence of such dyes (viewed under a dissecting microscope) allows us to discern whether injection is successfully performed during GONAD. For example, in our experiment, failure of GONAD is evident because leakage of blue solution from the oviduct is visible through observation under a dissecting microscope. When GONAD is carried out by a skilled person, such failure seldom occurs.

  1. Does and how the “Cas 9 leakage” affect the development of the embryo of the maternal organism, also changed by in utero electroporation?

Answer 5: As mentioned above, “Cas9 leakage” mentioned by the reviewer is caused by failure in GONAD. In this case, injection pipette might have been penetrated out through the oviductal wall or the solution might have been accidentally expelled before insertion of the pipette into the lumen of an oviduct. Notably, in our case, if both failures occur, no appreciable damages towards the development of preimplantation embryos are discernible. Furthermore, in vivo electroporation under the electric condition used here is not harmful to the development of those embryos. One of the factors affecting the development of early embryos is the strength of electric pulses generated from an electroporator. As voltage increases, it is obvious that frequent embryonic death occurs. For instance, our electric condition used for genome editing in B6C3F1 hybrid mice is not suitable for GONAD using C57BL/6 strain: this strain requires milder (less stringent) condition (Gurumurthy et al., Nature Protocols 14(8):2452-2482, 2019. doi: 10.1038/s41596-019-0187-x). Furthermore, we are now preparing to submit a paper (entitled, “Modification of i-GONAD suitable for production of genome-edited C57BL/6 inbred mouse strain” by Takabayashi et al.) to Journal “Cells”, in which C57BL/6 zygotes can be efficiently genome edited in situ by an electroporator used widely.

Similar events should occur upon in utero electroporation. Thus, careful examination for exploring optimal electric condition may be required prior to in utero electroporation.

  1. Could authors provide the data of intact genomic loci where theoretically could be observed off-target Cas9 activity besides the overall normal embryo morphology?

Answer 6: We have already looked into off-target activity issues that may be caused by CRISPR/Cas9 system. For example, there was no off-target mutation in the candidate genes (that are theoretically prone to be genome edited) when we attempted to destroy GGTA1 via CRISPR/Cas9 system (Sato et al., Xenotransplantation 2014, 21, 291–300. DOI: 10.1111/xen.12089). A similar case was also reported by Horii et al. (Sci. Rep. 2017, 7, 7891. doi.org/10.1038/s41598-017-08496-8) who checked possible off-target activity in the target gene Mecp-2, and found no mutations in the related genes.

  1. Could the number of nonspecific modification rise since the time/concentration dependence of Cas9-RNP occurrence in the embryo through the development stages?

Answer 7: As the half-life of Cas-RNP seems to be very short, it is conceivable that the Cas9-RNP introduced inside the zygotes will disappear at least by the blastocyst stages (3-4 days after fertilization). The presence of remnant amount of Cas-RNP may be the cause for generating mosaicism, which may occur frequently at 2-cell or later stage embryos. In our cases, 10-20% of embryos are those with mosaicism, even when i-GONAD is performed at late 1-cell stage. In case of i-GONAD, higher amounts of Cas-RNP (probably 10 to 100-fold higher amounts of Cas-RNP for zygote microinjection- or in vitro electroporation-based genome-edited mice production) are required. However, such high amounts of Cas-RNP do not appear to affect the development of preimplantation embryos. As mentioned previously, one of the critical factors affecting the development of early embryos upon i-GONAD is the electronic condition provided from an electroporator.

  1. How many days does the Cas9 exist in embryo? Is it possible to measure the titer of Cas9 after 0.7 to 1.7 day transition?

Answer 8: As mentioned above, persistent presence of Cas9 appears to be by the blastocyst stage. Our collaborator Dr. Takayuki Sakurai at Shinshu University produced transgenic mouse line overexpressing Cas9 systemically and observed no obvious abnormality in this line (Sakurai et al., Scientific Reports 6, 20011, 2016. DOI:10.1038/srep20011). These mice are used as those allowing genome editing at multiple target loci (Sakurai et al., Scientific Reports, 10, 1091, 2020. doi: 10.1038/s41598-020-57996-7). According to Dr. Sakurai, there are some commercially available anti-Cas9 antibodies, but all of these are unable to be used for both immunocytochemical staining and Western blotting. Thus, at present it seems difficult to measure the level of Cas9 introduced into early mouse embryos.

  1. Which characteristics of embryo were selected to provide it integrity?

Answer 9: As shown in the present text, we attempted to knock out GGTA1, whose product is involved in synthesis of cell-surface carbohydrate antigen, alpha-Gal epitope. The presence of alpha-Gal epitope can be detected by staining with fluorescence-labelled BS-I-B4 lectin. If GGTA1 is successfully targeted in mouse zygotes, they should be recognized as those that are negative for staining with the lectin. Since this staining is applicable to living embryos, it is possible to collect these lectin-negative embryos and transfer them to the uterine horn of pseudopregnant recipient female mice to allow further development in vivo. Consequently, the resultant fetuses should be genome edited. Unfortunately, KO of GGTA1 in mice did not cause loss of alpha-Gal epitope in this experiment. Due to this failure, we had to seek evidence for genome editing at the target locus through direct sequencing, due to which embryonic development was discontinued.

  1. Will the development of such an embryo be investigated further?

Answer 10: It may be desirable to add a noninvasive genetic trait such as EGFP into the genome of an embryo via targeted knock-in of an EGFP expression cassette. This means that in the embryos showing successful knock-in, EGFP expression (controlled by the upstream promoter of a target gene) will occur, but expression of the target gene will be blocked. This event can be realized in vivo when i-GONAD is employed. Expression of EGFP in the embryos (morulae or blastocysts) can be easily checked in vitro when they are isolated from the oviducts of i-GONAD-treated female. If only the fluorescent embryos are selected and immediately transferred to the reproductive tract of a recipient female, it would be possible to trace the developmental process of the genome-edited embryos in vivo.